# The Emergence of Objectness:
# Learning Zero-Shot Segmentation from Videos

**Runtao Liu**[1,2*]  **Zhirong Wu**[1*]  **Stella X. Yu**[3]  **Stephen Lin**[1]

Microsoft Research Asia[1]  John Hopkins University[2]  UC Berkeley / ICSI[3]
runtao219@gmail.com  stellayu@berkeley.edu  {wuzhiron,stevelin}@microsoft.com

## Abstract

Humans can easily segment moving objects without knowing what they are. That objectness could emerge from continuous visual observations motivates us to model segmentation and movement concurrently from *unlabeled* videos. Our premise is that a video contains different views of the same scene related by moving components, and the right region segmentation and region flow allow view synthesis which can be checked on the data itself without any external supervision.

Our model first deconstructs video frames in two separate pathways: an appearance pathway that outputs feature-based region segmentation for a single image, and a motion pathway that outputs motion features for a pair of images. It then binds them in a conjoint region flow feature representation and predicts *segment flow* that provides a gross characterization of moving regions for the entire scene. By training the model to minimize view synthesis errors based on segment flow, our appearance and motion pathways learn region segmentation and flow estimation automatically without building them up from low-level edges or optical flow respectively.

Our model demonstrates the surprising emergence of objectness in the appearance pathway, surpassing prior works on **1**) zero-shot object segmentation from a single image, **2**) moving object segmentation from a video with unsupervised test-time adaptation, and **3**) semantic image segmentation with supervised fine-tuning. Our work is the first truly end-to-end learned zero-shot object segmentation model from unlabeled videos. It not only develops generic objectness for segmentation and tracking, but also outperforms image-based contrastive representation learning without augmentation engineering.

## 1 Introduction

Contrastive learning [1–3] has recently become a powerful method for obtaining high-level visual representations from images [4]. While these representations are shown to be more generalizing, there remain two practical limitations: **1**) Hand-crafted augmentations such as image cropping and color jittering [5] are critical for achieving invariant recognition, and yet they fall short of capturing more complex object deformations and 3D viewpoint changes. **2**) Additional labeled data are required at the downstream for representation fine-tuning, preventing standalone applications.

Our goal here is to overcome these limitations of contrastive representation learning by developing object segmentation models automatically from unlabeled videos without any supervision. Unlike single static images, videos contain sequences of dynamic images that could reveal not only moving objects from their backgrounds, but also their internal part organizations with articulated movements. Once figure-ground segregation occurs automatically in raw videos, object semantics can be readily discovered from those foreground segmentations.

---

*Equal contribution. Work done when Runtao was a StarBridge intern at MSRA.

35th Conference on Neural Information Processing Systems (NeurIPS 2021).

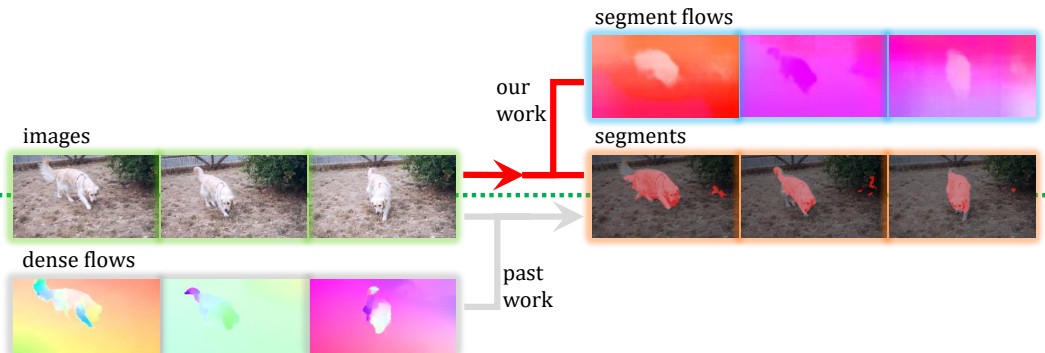

Figure 1: Our zero-shot object segmentation is learned from an unsupervised factorization of images into segments and their motions, whereas past work segments objects based on dense pixel-wise optical flows, which are brittle in the presence of noise, articulated movement, and abrupt motion.

Several observations motivate us to explore such zero-shot learning of object segmentation. **1)** Humans can easily segment moving objects without knowing what they are. **2)** In biological vision, newborn chicks raised in a controlled visual world rapidly develop more accurate object representations when presented with temporally slow and smooth objects, generalizing from very limited viewpoints [6, 7]. **3)** Invariant recognition can be developed by seeking slowly varying features from temporally varying signals [8], disentangling object identity and object location unsupervisedly.

We model segmentation and movement concurrently from *unlabeled* videos. Our premise is that a video contains different views of the same scene related by moving components, and the right region segmentation and region flow would allow mutual view synthesis between frames that can be checked on the data itself. That is, if we know how regions of frame $j$ are moved from regions of frame $i$, we can synthesize frame $j$ by copying regions from frame $i$ and paste them according to how they move. Comparing the synthesized frame $j$ with the actual frame $j$ provides feedback on how to improve both region segmentation and region flow estimation without needing any supervision.

View synthesis has been frequently adopted as a self-supervised criterion for learning dense optical flows [9], monocular depth [10], and multi-plane image representation [11] etc from images. Unlike prior works that focus on low-level visual correspondences, our work tackles object segmentation for mid- to high-level visual recognition directly. Specifically, as illustrated in Figure 1, instead of deriving dense optical flows between successive frames and supplying additional cues for image-based object segmentation in a bottom-up manner, we seek a top-down factorized representation that provides a gross characterization of moving regions for the entire scene.

Our model first deconstructs video frames by processing them in two separate pathways: an appearance pathway that models *what is moving* and outputs feature-based region segmentation given a single image, and a motion pathway that models *how it moves* and outputs motion features given a pair of images. It then binds them in a conjoint region flow feature representation, based on which we predict *segment flow* as the *common fate* [12] or *piecewise constant movement* of all the pixels in the same region. By training the model to minimize view synthesis errors based on segment flow, our appearance and motion pathways learn region segmentation and flow estimation automatically without building them up from low-level edges or optical flows respectively.

After training our segmentation and flow features for view synthesis on unlabeled videos, our model demonstrates the surprising emergence of *objectness* in a particular feature channel of the appearance pathway. That is, our model can be directly applied to novel images and videos for segmenting foreground objects: Our appearance pathway can perform zero-shot object segmentation on a single image, whereas our overall model can perform zero-shot moving object segmentation on a single video with test-time adaptation. Our image feature learned from unlabeled videos can be further fine-tuned on a small labeled dataset for semantic segmentation. Experimentally, we demonstrate strong performance on all three applications, with considerable gains over baselines.

To summarize, our work makes the following contributions. **1)** We develop the first truly end-to-end learned zero-shot object segmentation model from unlabeled videos, assuming no low-level computation such as edges or optical flow. **2)** We bypass the traditional low-level dense optical flow and propose to compute novel mid-level segment flow directly. **3)** Our model not only develops

generic objectness for segmentation and tracking, but also outperforms prevalent image-based contrastive learning methods without augmentation engineering. Our code is available at `https://github.com/rt219/The-Emergence-of-Objectness`.

## 2 Related Works

**Video object segmentation.** Segmentation of moving objects requires finding correspondences across time. A major line of work assumes that an object mask is given in the initial frame, and the goal is to propagate the mask to future frames based on the similarity of learned visual representation. Such a representation can be trained from pixel-level object masks with long-term relations in videos [13, 14], or from self-supervised criteria such as colorization [15] and cycle-consistency [16].

Fully unsupervised video object segmentation *without initial masks* has received less attention. NLC [17] and ARP [18] segment moving objects based on temporal clustering, but they rely on edge and saliency annotations, and are thus not completely unsupervised. FTS [19] performs segmentation by obtaining motion boundaries from optical flow. SAGE [20] takes into account edges, motion segmentation, and image saliency for video object segmentation. Contextual information separation [21] segments moving objects by exploiting independence between the foreground motion and the background motion. A concurrent work based on motion grouping [22] clusters pixels by similar motion vectors. These works rely on off-the-shelf optical flow results, which may be trained with [23, 24] or without [9] supervision. Our work does not assume any known low-level features such as edges or optical flow, and learns the right feature to extract completely from scratch.

**Motion Segmentation.** Classical methods for motion segmentation [25–27] find regions of distinctive motion based on two-frame optical flow. Supervised learning approaches [28, 29] map optical flow to segmentation masks. Ideally, these methods require dense and accurate optical flow. In practice, the low-level differential flow is often present along edges; it is neither dense nor smooth within a region, often inhomogeneous for deformable and articulated objects [30], and sensitive to varying scene depths and camera motion [31–33, 33]. Motion segmentation is shown to be less brittle when examined over a large time interval [30]. Trajectory clustering [34] tracks point trajectories over hundreds of frames, extracts descriptors for the trajectories, and clusters them to obtain a segmentation. Such a global approach is computationally demanding.

In contrast, our segmentation is based not on motion between two frames, but on image appearance in a single image, which provides rich cues such as color, texture, and edges for pixel grouping and segregation. While our segmentation model does not need dense pixel correspondences between frames, it is learned to be in sync with region-wise correspondences for best view synthesis.

**Unsupervised learning for segmentation.** Human annotation of pixel-wise segmentations is not only time-consuming, but also often inaccurate along boundaries. Learning segmentation without labels is thus of great interest in practice. SegSort [35] predicts segmentation by learning to group super-pixels of similar appearance and semantic context from static images. Later work [36] contrasts holistic mask proposals obtained by traditional bottom-up grouping.

A related line of work focuses on learning part segmentation from images and videos of the same object category, such as humans and faces. SCOPS [37] is a representative co-part segmentation method, learned in a self-supervised fashion; its general idea follows unsupervised landmark detection [38], leveraging geometric invariance, representation equivariance, and perceptual reconstruction. [39] explores motion cues in videos to discover object part organization and dynamics. Motion-supervised co-part segmentation [40] models part motion between adjacent frames using affine parameters. A similar idea is implemented with capsule networks [41]. In contrast, our work is not restricted to videos in a *single* category and learns object segmentation from a collection of generic videos.

**Learning objectness from data.** Segregating foreground objects from background is a central problem in visual recognition. Prior works on images first generate hierarchical segmentations [42–44] using low-level visual cues such as colors and boundaries [45], and then rank these candidate regions according to a certain criterion [46, 47]. These approaches often generate many overlapping redundant individual object instance proposals.

Videos of slowly moving objects [8] are shown to enable development of newborn vision [6, 7]. Linear models [48, 49] can factorize a sequence of images into foreground and background layers,

assuming independent motion among them. Layered representations are also used for optical flow estimation [50, 27, 51], view-interpolation, and time retargeting [52–55].

Our work adopts an unordered layered representation with multiple segmentation channels, each identifying a region of common motion. While our view synthesis objective during training does not differentiate foreground or background in these channels, surprisingly, *objectness* emerges automatically in a particular channel after we train our model from a collection of raw videos.

**Image representation learning from videos.** Motion reveals the location, shape, and part hierarchy of moving objects. Motion segmentation has thus been used to supervise learning of image-level object representations [56]. Motion propagation [57] predicts dense optical flow from sparse optical flow, conditioned on the color image. Unlike prior works, the single image representation produced by our model is not learned *from* motion supervision, but concurrently learned *with* between-frame region flow as a by-product of our moving object segmentation from unlabeled videos.

## 3   Segmentation by Appearance-Motion Decomposition

We would like our model to segment moving objects without necessarily knowing what and how many they are. Our model is trained on a collection of unlabeled generic videos, and can be directly deployed on a novel image (video) to produce (moving) object segmentation. No human annotations are required during either training or testing.

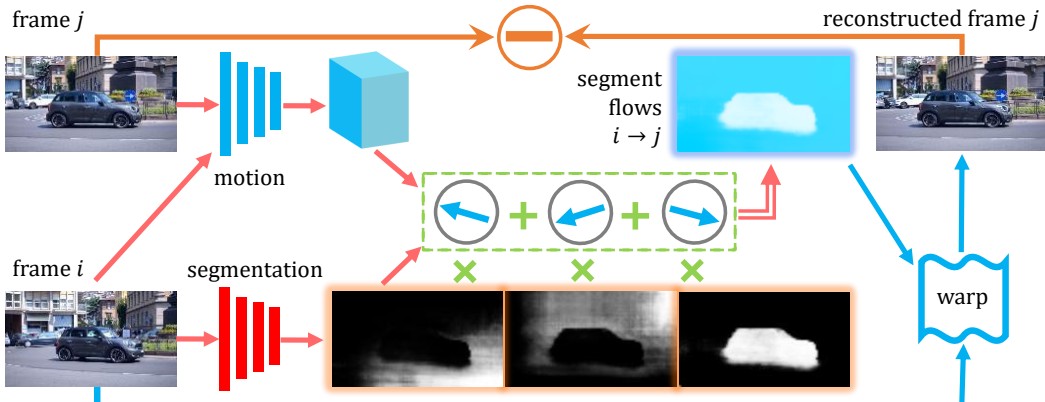

Figure 2: We learn a single-image segmentation network and a dual-frame motion network with an unsupervised image reconstruction loss. We sample two frames, $i$ and $j$, from a video. Frame $i$ goes through the segmentation network and outputs a set of masks, whereas frames $i$ and $j$ go through the motion network and output a feature map. The feature is pooled per mask and a flow is predicted. All the segments and their flows are combined into a segment flow representation from frame $i \rightarrow j$, which are used to warp frame $i$ into $j$, and compared against frame $j$ to train the two networks.

Illustrated in Figure 2, our so-called *appearance-motion decomposition* (AMD) model first deconstructs a pair of video frames, $X_i$ and $X_j$, in two separate pathways. The bottom appearance pathway $f_A$ takes in a single image $X_i$ and outputs a feature-based segmentation, whereas the top motion pathway $f_M$ takes in both frames $(X_i, X_j)$ and outputs the flow features between them. The two pathways then come together to construct a conjoined segment flow representation $F$, which is used to synthesize frame $X_j$ by warping $X_i$. The overall model can be trained to minimize the reconstruction error on frame $X_j$ over sampled image pairs in the training dataset.

**1) Appearance pathway for feature-based segmentation.** We adopt a fully convolutional neural network for segmenting a single RGB image of size $h \times w$ into $c$ regions, where $c$ is a hyper-parameter. Formally, video frame $X_i \in \mathbb{R}^{3 \times h \times w}$ is transformed by $f_A$ into segmentation $S$ with $c$ soft masks:

$$S = f_A(X_i) \in \mathbb{R}^{c \times h \times w}, \tag{1}$$

$$\sum_{m=1}^{c} S(p) = 1, \qquad p = 1, \dots, h \times w. \tag{2}$$

The above normalization equation reflects that values of $S$ are the probabilities of pixel $p$ belonging to $c$ regions. Empirically, we choose $c=5$ by default. Our ablation study later shows that a larger $c$ may lead to over-segmentation, whereas a smaller $c$ may lack the spatial resolution to locate objects.

Note that our segmentation network is based on the RGB appearance in a single frame instead of optical flow between two frames. Since it is designed to operate on static images, it can be transferred to downstream image-based vision tasks. Our segmentation network trained on unlabeled videos can be directly used to segment not only moving objects from novel videos (Section 4.2), but also salient objects from single images in a zero-shot fashion (Section 4.1). It can be further fine-tuned on a labeled image dataset for semantic segmentation (Section 4.3).

**2) Motion pathway for correspondences.** We adopt PWC-Net [23] for extracting pixel-wise motion features between a pair of images. PWC-Net is originally designed for predicting dense optical flow, and the feature for each pixel in one frame captures its similarity to spatial neighbors in the other frame. Formally, video frames $(X_i, X_j)$ are transformed by $f_M$ into motion correspondence feature $V$ of $d_v$ dimensions:

$$V = f_M(X_i, X_j) \in \mathbb{R}^{d_v \times h \times w}. \tag{3}$$

Note that we only adopt the network architecture *not* the trained weights of PWC-Net in [23], and our choice among alternative architectures such as FlowNet [58], FlowNet2 [59], SpyNet [60], and RAFT [24] is based on conceptual simplicity and light-weight model size.

**3) Segment flow representation.** We now construct a conjoined segment flow representation from both pathways to enable view synthesis. Specifically, we pool the pixel-wise correspondence feature $V$ in the motion pathway according to the image segmentation $S$ in the appearance pathway, resulting in an average $d_v$-dimensional motion feature per segment, i.e., $V_m$ for the $m$-th mask $S_m$:

$$V_m = \frac{\sum_{p=1}^{h \times w} V(p) \times S_m(p)}{\sum_{p=1}^{h \times w} S_m(p)} \in \mathbb{R}^{d_v}, \quad m=1,\ldots,c. \tag{4}$$

We then predict a single common 2D flow vector $F_m$ for the entire $m$-th segment $S_m$ based on its average motion feature $V_m$, using a two-layer multilayer perceptron (MLP) for the head function $g$:

$$F_m = g(V_m) \in \mathbb{R}^2, \quad m=1,\ldots,c. \tag{5}$$

So far, we deconstruct a pair of video frames $(X_i, X_j)$ into $c$ segmentation masks and their associated flow vectors $\{(S_m, F_m) : m = 1, \ldots, m\}$, assuming one common motion for pixels within the same segment. This piece-wise constant motion assumption simplifies flow estimation and provides a gross characterization of movement in the scene. While it may not hold for deformable and articulated objects, when trained over a collection of videos, our model with appearance feature-based segmentation is able to aggregate smoothly moving pieces in a wholesome segment.

We then compose these moving components into a novel flow representation $F$ for the entire image:

$$F(p) = \sum_{m=1}^{c} F_m \times S_m(p), \quad p=1,\ldots,h \times w. \tag{6}$$

We call $F$ *segment flow*, as its values indicate the overall displacement at the segment level. This conjoined representation allows motion and segmentation to cross-supervise each other. Given a between-frame flow vector, the segmentation network could be supervised to find pixels of this offset. Given a segmentation mask, the motion network could be supervised to find the flow for this mask.

Our approach to image segmentation with motion inputs is fundamentally different from motion segmentation methods: **1)** Our segmentation mask is predicted from static image appearance that does not require dense and accurate flow for supervision; **2)** Our flow estimation is at the segment level, which can be inferred from sparse and noisy pixel-level flow estimates.

**4) Reconstruction objective for view synthesis.** How do we validate our segment flow $F$, a conjoined representation from both appearance and motion pathways? Intuitively, the right segmentation $S$ and motion $F_m$ would allow the synthesis of frame $X_j$ from frame $X_i$ according to their segment flow $F$, and the reconstruction $\hat{X}_j$ should be close to the actual frame $X_j$:

$$\hat{X}_j(p) = X_i(p + F(p)), \quad p = 1,\ldots,h \times w \tag{7}$$

$$\mathcal{L} = D(X_j, \hat{X}_j), \tag{8}$$

where $D$ is a metric measuring the distance between two images. We adopt the pixel-wise photometric loss SSIM [61] for simplicity, among alternatives such as deep-feature matching losses [62, 63] and contrastive losses [64]. The warping loss $\mathcal{L}$ is the only self-supervision our model receives.

Note that a small reconstruction error does not necessarily mean that the segmentation and flow are correct, but correct segmentation and flow must result in a small reconstruction error. That is, this reconstruction objective is a necessary condition for correct segmentation and flow estimation.

Compared to traditional optical flow estimation, our model also derives motion flow from brightness constancy between two frames, but the optical flow computation assumes pixel-wise local displacements that are independent of each other, whereas our segment flow assumes all the pixels in the same segment have a common displacement. In addition, our segmentation is determined from the image appearance, instead of fixed-size patches assumed in the Lucas–Kanade optical flow method [65].

Our model has thus two essential bottlenecks: One is the number of segments, and the other is piecewise constant segment flow. Both are important for directly delivering a mid-level organization without building it up from low-level vision such as edges and optical flow.

**5) The emergence of objectness.** The appearance pathway in our model only segments an image into $c$ regions, one in each of its $c$ channels. Our reconstruction objective is only concerned with collective view synthesis from these $c$ channels, and does not designate any channels for moving objects or background. That is, any channel could contain the moving foreground object for a particular video.

Surprisingly, we observe empirically that objects in training videos tend to concentrate in the same channel, with a relatively sharp and uniform mask in the center of the image (Figure 3). We conduct analysis to understand the emergence of objectness in a particular channel, the channel activated by the feature that seems to capture generic objects against their backgrounds.

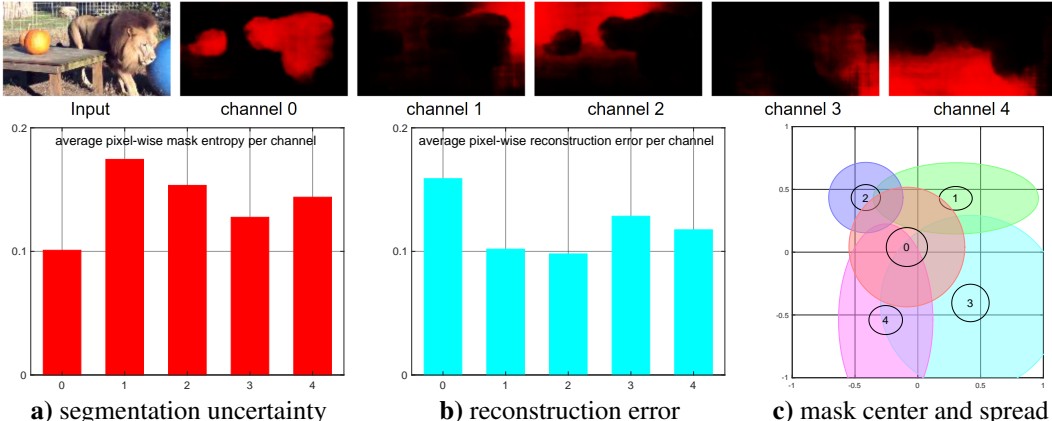

Figure 3: **Top)** Our model shows the surprising emergence of objectness in a particular channel. Note that the index of the channel (channel 0 here) could be random in different training runs, but there is always a channel concentrated with objects from all the training videos. **Bottom)** Channel-wise statistics over 17,500 sample training frames of our segmentation network reveal that our objectness channel has **a)** the least segmentation uncertainty (measured by the entropy of $S_m$), **b)** the largest reconstruction training error (measured by SSIM), and **c)** mostly central locations (the average of the mean and standard deviation of mask centers marked by the channel number and the small black circle) and relatively focused areas (the half of average mask spread shown as the color-shaded disk).

We calculate three types of per-pixel statistics for each segmentation channel from 17,500 sampled frames across training videos: **1)** segmentation uncertainty measured by the entropy of mask value $S_m$, **2)** reconstruction error between $X_j$ and $\hat{X}_j$ measured by SSIM, and **3)** the mean and standard deviation of the mask center and the average mask spread. Figure 3 shows that the objectness channel has the least segmentation uncertainty, the largest reconstruction training error, mostly central locations and relatively focused areas.

Our conjecture is that three factors contribute to the emergence of objectness in our model: **1)** Training videos tend to track moving objects in the center field of view; **2)** Our piece-wise constant motion assumption holds better for the background; **3)** Motion of object pixels tends to be independent of

that of background pixels, whereas motion of background pixels could be interpolated from other pixels scattered in multiple background regions.

**6) Limitations.** Our current model develops a general sense of objectness from a collection of unlabeled videos. It can only segment foreground objects without differentiating between object instances or between semantic classes. It is not guaranteed to segment out all the objects or full objects. Like most data-driven learning methods, our model performance heavily depends on the properties of training videos, the coverage of object categories, and the object motion as well as camera motion. Nevertheless, we still find it amazing that our model is able to generalize the concept of objectness across a variety of datasets in a zero-shot fashion, moving a step closer to human vision.

## 4  Experiments

**Tasks.** We train our AMD model on unlabeled videos and test it on three downstream applications. **1) Zero-shot object segmentation.** We directly apply our segmentation network to static images for salient object detection. **2) Zero-shot moving object segmentation.** We apply our AMD model to segment moving objects in novel videos with zero human labels. **3) Fine-tuning for semantic segmentation.** We fine-tune our appearance pathway on labeled images for semantic segmentation.

**Training data.** Our training videos come from Youtube-VOS [66], a large object-centric video dataset. Its training split contains about 4,000 videos covering 94 categories of objects. The total duration of the dataset is 334 minutes. We sample video frames at 24 frames per second, without using any segmentation labels provided in Youtube-VOS.

**Implementation details.** We train our model from scratch without external pretraining. For the segmentation network, we use ResNet50 [67] as our backbone followed by a fully convolutional head containing two convolutional blocks. For the motion network, we use PWC-Net [23]. We resize the shorter edge of the input image to 400 pixels, and randomly crop a square image of size $384 \times 384$ with random horizontal flipping augmentation. No other augmentations are used. We adopt the symmetric reconstruction loss that considers either frame as the target frame and sums the two reconstruction errors. We use the Adam optimizer with a learning rate of $1 \times 10^{-4}$ and a weight decay of $1 \times 10^{-6}$. We train AMD on $8\times$ V100 GPUs, with each processing two pairs of sampled adjacent frames. The network is optimized for 400K iterations.

### 4.1  Zero-Shot Saliency Detection

We directly evaluate our Youtube-VOS trained segmentation network on **DUTS [74]**, a salient object detection benchmark which contains 5,019 test images with pixel-level ground truth annotations. We follow two widely used metrics: the $F_\beta$ score and the per-pixel mean squared errors (MAE). $F_\beta$ is defined as the weighted harmonic mean of the precision ($P$) and recall ($R$) scores: $F_\beta = \frac{(1+\beta^2)P \times R}{\beta^2 P + R}$, with $\beta^2 = 0.3$. MAE is simply the per-pixel averaged error of the soft prediction scores.

**Experimental results.** We compare our saliency estimation results against several traditional methods based on low-level cues and various priors: background priors [68], objectness [70, 72], and color contrast [75]. Table 1 shows that our method achieves an $F_\beta$ score 60.2 and an MAE score of 0.13, outperforming baselines by sizable margins. Note that AMD is not designed specifically for this task or this dataset, and its strong performance demonstrates the generalization power of our model.

Table 1: Salient object detection performance on the DUTS dataset. Our model outperforms traditional low-level methods by notable margins.

| Model | $F_\beta$ | MAE |
|---|---|---|
| RBD[68] | 51.0 | 0.20 |
| HS[69] | 52.1 | 0.23 |
| MC[70] | 52.9 | 0.19 |
| DSR[71] | 55.8 | 0.14 |
| DRFI[72] | 55.2 | 0.15 |
| **AMD** | **60.2** | **0.13** |

Table 2: Transfer performance for semantic segmentation on VOC2012. Our method outperforms TimeCycle and compares favorably with contrastive methods.

| Model | Data | Aug. | mIoU |
|---|---|---|---|
| Scratch | – | – | 48.0 |
| TimeCyle[73] | VLOG | light | 52.8 |
| MoCo-v2[2] | YTB | light | 61.5 |
| **AMD** | YTB | light | **62.0** |
| MoCo-v2[2] | YTB | heavy | **62.8** |
| **AMD** | YTB | heavy | 62.1 |
| MoCo-v2[2] | IMN | heavy | **72.4** |

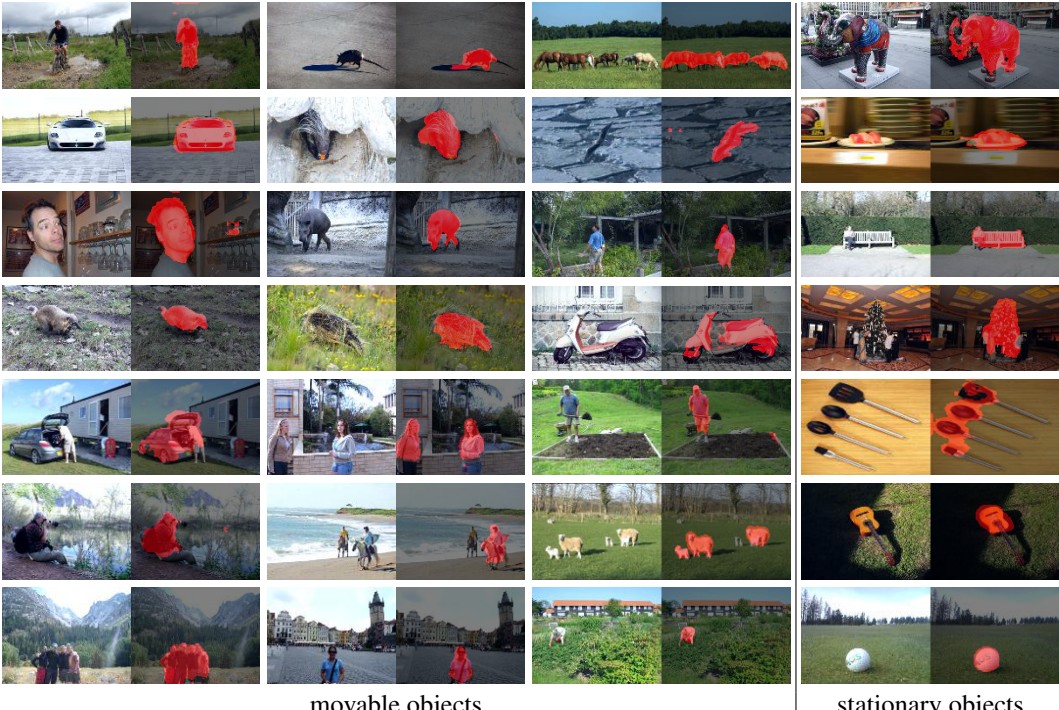

movable objects    |    stationary objects

Figure 4: Sample salient object detection results. We directly apply our pretrained segmentation network to novel images in DUTS without any finetuning. Surprisingly, we find that the model pretrained on videos to segment moving objects can generalize to detect stationary unmovable objects in a static image, e.g. the statue, the plate, the bench and the tree in the last column.

In related unsupervised learning of saliency detection [76–78], the priors of traditional low-level methods are ensembled. Though they do not use saliency annotations, their models are pretrained for ImageNet classification and even semantic segmentation with pixel-level annotations. These methods are thus not fully unsupervised and omitted from comparisons.

Figure 4 shows sample results on salient object detection. Surprisingly, we find that our model trained on Youtube-VOS to segment moving objects not only detects movable objects in single images, but also stationary unmovable objects such as statues, benches, trees and plates. These results suggest that our model learns generic objectness from unlabeled videos.

To quantify this observation, we manually label objects from the DUTS dataset into *movable objects* and *stationary objects*. The F1 score of movable objects and stationary objects are 63.0 and 57.9 respectively, without a significant performance gap. We hypothesize that our model could also learn objectness from camera motion, which causes objects and backgrounds at various depths to have different 2D optical flow even though the objects are static.

## 4.2 Zero-shot Video Object Segmentation

Since our method does not require any labels, we apply our AMD model to object segmentation in novel videos using test-time adaptation: Given a novel video, we optimize the training objective in Eq. 8 on pairs of frames sampled from the test video. The adaptation takes 100 iterations per video.

We evaluate zero-shot video object segmentation on three datasets. **DAVIS 2016 [79]** contains 20 validation videos with 1,376 annotated frames. **SegTrackv2 [80]** contains 14 videos with 976 annotated frames. Following prior works, we combine multiple foreground objects in the annotation into a single object for evaluation. **FBMS59 [30]** contains 59 videos with 720 annotated frames. The dataset is challenging as the object may be static for a period of time. We pre-process ground-truth labels as in [21]. For evaluation, we report the Jaccard score, which is equivalent to the intersection over union (IoU) between the prediction and the ground truth segmentation.

Table 3: Unsupervised video object segmentation performance on DAVIS 2016, SegTrackv2 and FBMS59 datasets, measured in terms of Jaccard score. The table is split into traditional non-learning-based and recent self-supervised learning methods. Results which rely on other kinds of human supervisions (Sup.) are grayed. Dependence for pretrained dense flow method is also listed for each model. MG's results on SegTrackv2 and FMBS59 using ARFlow are reproduced by ours and marked with ∗. We evaluate AMD in two settings: appearance pathway only and both pathways with test time adaptation. AMD performs favorably to CIS on DAVIS 2016, while showing large gains on the other two benchmarks.

| | Model | e2e | Sup. | Flow | DAVIS 2016 | SegTrackv2 | FBMS59 |
|---|---|---|---|---|---|---|---|
| traditional | SAGE[81] | ✗ | ✗ | LDOF[82] | 42.6 | 57.6 | 61.2 |
| | NLC[17] | ✗ | edge | SIFTFlow[83] | 55.1 | 67.2 | 51.5 |
| | CUT[34] | ✗ | ✗ | LDOF[82] | 55.2 | 54.3 | 57.2 |
| | FTS[19] | ✗ | ✗ | LDOF[84] | 55.8 | 47.8 | 47.7 |
| | ARP[18] | ✗ | saliency | CPMFlow[85] | 76.2 | 57.2 | 59.8 |
| learning | CIS[21] | ✗ | ✗ | PWC[23] | **59.2** | 45.6 | 36.8 |
| | MG[22] | ✗ | ✗ | ARFlow[9] | 53.2 | 37.8∗ | **50.4**∗ |
| | **AMD** (per-img) | ✓ | ✗ | ✗ | 45.7 | 28.7 | 42.9 |
| | **AMD** (per-vid) | ✓ | ✗ | ✗ | 57.8 | **57.0** | 47.5 |

**Experimental results.** We consider baseline methods claiming to be unsupervised for the full pipeline: traditional non-learning-based approaches [81, 17, 19, 34, 18] and recent self-supervised learning methods [21, 22]. Table 3 summarizes results for all the methods on the three datasets. Note that NLC [17] actually relies on an edge model trained with human-annotated boundaries, whereas ARP [18] depends on a segmentation model trained on a human-annotated saliency dataset. We thus shade their entries in gray. For all the traditional methods, since the original papers do not report results on most of these benchmarks, we simply provide their performance reported in CIS [21].

We evaluate AMD with and without test-time adaptation. No adaptation boils down to per-image saliency estimation using only the appearance pathway, whereas adaptation fine-tunes both appearance and motion pathways. On DAVIS 2016, our method achieves a Jaccard score of $57.8\%$, surpassing all traditional unsupervised models. For CIS [21], their best performing model uses a significant amount of post-processing, including model ensembling, multi-crop, temporal and spatial smoothing. We thus refer to their performance obtained from a single model without post-processing. Our model is slightly worse than CIS on DAVIS, by $1.4\%$. However, on SegTrackv2 and FBMS59, our method outperforms CIS by large margins of $11.4\%$ and $10.7\%$ respectively. Motion grouping [22] is a work concurrent with ours. It is a motion segmentation approach that relies on an off-the-shelf pre-computed dense optical flow model. Motion grouping performs worse than our method on DAVIS2016 and SegTrackv2 when a low-performance unsupervised optical flow model ARFlow is used [9]. With a state-of-the-art supervised optical flow model [24] which is trained on ground truth flow, their performance improves significantly. Among all the discussed methods, ours is the first end-to-end self-supervised learning approach which does not require a pretrained optical flow model.

Figure 5 shows result comparisons with CIS [21]. For most of these examples, our segment flow only coarsely reflects the true pixel-level optical flow. However, our segmentation results are significantly better and less noisy, insensitive to the flow quality. In the first and the third examples, our model produces high-quality object segmentations even though the object motion cues are weak.

### 4.3 Semantic Segmentation

Since our Youtube-VOS trained segmentation network can already segment generic objects, we further examine its modeling power of semantic segmentation on **Pascal VOC 2012 [86]**, which contains 20 object categories with 10,582 training images and 1,449 validation images. We finetune our AMD model on the PASCAL VOC training set and evaluate it on the validation set. The finetuning takes 40,000 iterations with batch size 16 and the initial learning rate 0.01. The learning rate undergoes polynomial decay with a power parameter of 0.9.

**Experimental results.** Our baselines are an image-based contrastive model, MoCo-v2 [2], and a self-supervised video pretraining model, TimeCycle [73]. TimeCycle is pretrained on the VLOG

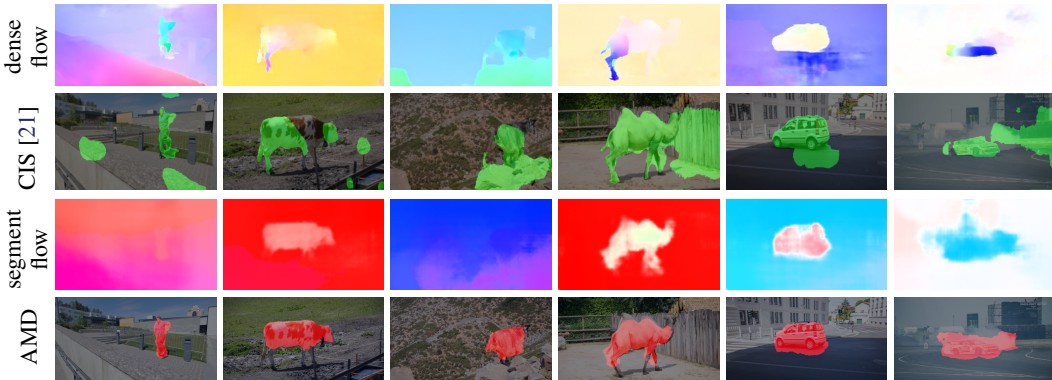

Figure 5: Comparisons with motion segmentation method CIS [21]. CIS segmentation is sensitive to noise, articulated motion, and camera motion in its dense flow. By disentangling appearance from motion, our AMD is less prone to these vulnerabilities, resulting in better and more robust results.

dataset, which is larger than our Youtube-VOS dataset. For MoCo-v2, we also pretrain the contrastive model on Youtube-VOS, to ablate the role of pretraining datasets. Since our method does not utilize heavy augmentations as in contrastive models, we also study the effects of data augmentations. Table 2 shows that our method outperforms the video pretrained TimeCycle significantly by 9.2%. With light augmentation (resizing, cropping), our model slightly outperforms MoCo-v2 by 0.5%. However, with heavy data augmentation (color jitter, grayscale, blurring), our method underperforms MoCo-v2 by 0.7%. The reason is that our model does not directly relate augmentations, and thus cannot build up invariance effectively across augmentations. MoCo-v2 performs much stronger when pretrained on ImageNet, possibly because the semantic distribution of ImageNet is well aligned with that of VOC2012. Overall, our model outperforms a prior self-supervised video model TimeCycle and compares favorably to a contrastive model MoCo-v2 under the same training data setting.

## 4.4 Ablation Study

The number of segmentation channels, $c$, is an important hyper-parameter of our model. Figure 6 shows our model predictions trained for $c = 5, 6, 8$; training becomes unstable when $c \leq 4$. A larger $c$ tends to lead to over-segmentation: The car and the swan are split into multiple regions even when the motion is very similar between separated regions. The model trained with $c = 5$ segments a full object, while the model trained with $c = 8$ separates the object into parts. Quantitatively, the video object segmentation performance on DAVIS2016 drops as we increase the number of segments.

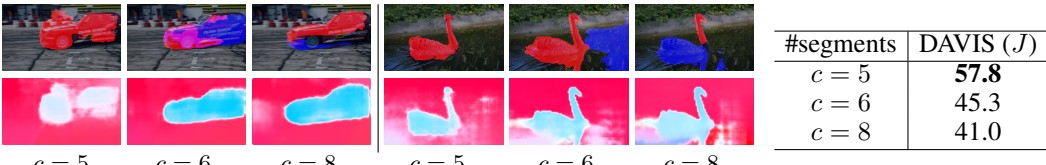

| #segments | DAVIS ($J$) |
|-----------|-------------|
| $c = 5$   | **57.8**    |
| $c = 6$   | 45.3        |
| $c = 8$   | 41.0        |

Figure 6: Ablation study on $c$ with different numbers of segments. **Left)** Two sample results with segmentation masks and segment flows. **Right)** Jaccard scores on DAVIS2016. As $c$ increases, the object region is oversegmented, decreasing the video object segmentation performance.

**Summary.** We show that objectness emerges from our AMD model trained on unlabeled videos. Our model first deconstructs video frames into appearance and motion, and then binds them into a conjoined segment flow representation for view synthesis. While prior works rely on accurate dense optical flow for object segmentation, our method learns from scratch on raw pixel observations. While our segment flow is a coarse characterization of motion, our object segmentation is in fact more robust. Validated on several segmentation benchmarks, our AMD model is the first end-to-end learning approach for zero-shot object segmentation without using any pretrained modules.

**Acknowledgements.** This work was supported, in part, by Berkeley Deep Drive to SY.

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
