# OpenReview forum: "The Emergence of Objectness: Learning Zero-shot Segmentation from Videos"
_NeurIPS.cc/2021/Conference — NeurIPS 2021 Poster_

### Official Review · Reviewer_zbkz · 2021-07-16

**Rating:** 5
**Confidence:** 4

**Summary:**

This work develops an approach to unsupervised single-object segmentation. The main idea is to segment the scene into $C$ (hyperparmeter) segments in one frame and to learn a flow field w.r.t. another frame such that it can be reconstructed by translating the segments from the original frame. This simple strategy provides competitive accuracy on standard object video segmentation benchmarks, as well as shows promise in unsupervised saliency detection (DUTS) and representation learning (VOC12).

**Limitations And Societal Impact:**

Some of the key model limitations do become apparent after reading the manuscript. Nevertheless, I would recommend offering this discussion more prominently in the main text.
Societal impact is not addressed in sufficient depth (Sec. 3 in supp. “Broader Impact” mentions only “privacy”), hence needs to be elaborated on more in the revision.

**Main Review:**

*Originality.*
The idea seems to be original and simple. In fact, so simple that a number of key design points remain unaddressed:
- the model is very sensitive to $C$, the number of masks for prediction, as becomes evident from Tab. 4.
- selecting the segmentation mask based on its proximity to the centre (l.184) does not appear principled, is highly dataset dependent and puts the fairness of the comparison to prior art into question (esp. for the saliency benchmark).
I also suspect that disocclusions (“holes”) and self-occlusions may pose a problem. In the first case, suppose an object moves and the camera is static. The parts of the image that become disoccluded cannot be reconstructed from the image where there were occluded in the previous frame. How does the method cope with it? For the second point, assume an object (e.g. a ball) rotating around its own axis. Approaches based on dense flow will be able to detect this motion. How would the method handle this scenario?

*Quality.*
The work is well-presented and argued. In particular, the argument that previous work relies on optical flow pre-training whereas the presented method doesn’t, is a good point. However, I would still not overstress it, since the method does use an optical flow network for motion prediction (PWC-Net); and dense optical flow can be learned from synthetic data.
The work also needs to further clarify the following points:
- What is the model accuracy without test-time adaptation? When is it reported with and without this process in the tables?
- How can it be that Fig. 3 shows significant improvements w.r.t. CIS, yet in Tab. 1 CIS actually performs a bit better?
- How is $C$ selected exactly? (l. 140 “empirically” is not concrete). This needs to be clear for every experiment and the dataset.
- What data (datasets) do the methods in Tab. 2 use for training / hyperparameter selection? This needs to be reported for transparency.

*Clarity.*
The work is generally well-written, but not without a few minor deficiencies:
- notation is not rigorous. For example, Eq. 5 multiples $\mathbb{R}^2$ with $\mathbb{R}^{c \times h \times w}$.
- some missing implementation details in Sec. 3.6: batch size, data augmentation used; how was the termination criterion selected?


*Significance.*
The approach provides tangible improvements w.r.t. previous work, is slightly worse than concurrent work ([6], but it is less of a concern). However, the method also appears much more prone to hyperparameter choice; how those were selected is important for the fairness of the comparisons and needs to be stated unambiguously.


**Post-rebuttal update**

I thank the authors for detailed clarifications. I find the presented results very encouraging in view of the method's simplicity and versatility. However, I still maintain that head-to-head comparisons to related methods (i.e. using the same training data) are essential, whenever technically possible, in order to understand where the approach fits in a larger body of previous work (and even if this use case was not originally intended). One of such experiments in the VOS setting -- pre-training on synthetic data -- is feasible, but not presented.

More generally relating to my point on (self-)occlusions, I'd also like to recommend including a short discussion about the assumptions underlying the approach mainly to clarify: When does it make sense to apply segment flow for the reconstruction loss? This is because such transformation generally does not apply due to the 3D nature of the scene.

Overall, I share the sentiment of Reviewer Y1LA that this work has potential, and encourage the authors to re-submit the manuscript after the revision.

**Time Spent Reviewing:**

3

---

> ### Author Response · Authors · 2021-08-10
> **To Reviewer zbkz**
>
> Thank you for the detailed comments.  These comments would help us greatly improve the quality of the paper.
>
> **The model is sensitive to $C$, and how is $C$ selected?**
>
> Like basically any model that has hyper-parameters, our model relies on an important hyper-parameter, $C$. We tune this hyper-parameter on the pretraining dataset and use a single pretrained model under the value $C=5$ for the downstream sets: DAVIS, SegTrack, FBMS and DUTS. Ablation of this hyper-parameter is provided in Table 4.
>
> **Selection of the object segmentation mask.**
>
>
> There seems to be a misunderstanding about the segmentation mask selection process. We determine the object segment channel (1 out of 5 channels) on the Youtube-VOS training videos. Once the object segment channel is chosen, it remains **fixed** for all testing videos and images, without per dataset or per instance adjustment. We empirically found that objects across datasets tend to occur in this single channel; this channel could also be easily chosen as the single consistent channel with maximal object motion.
>
> The model pretrained on Youtube-VOS is **directly** tested on DUTS saliency detection testing set, **without tuning a single parameter.** Therefore, the comparison with the baselines is fair.
>
> **Disocclusion and self-occlusion problems.**
>
> Dense optical flow methods handle the occlusion problem by estimating an occlusion map from the backward flow. This could be applied to our segment flow as well, by utilizing the occlusion map estimated from the backward flow for supervising forward flows.  We did not explicitly consider occlusion in the current paper to keep our idea simple and our main task in focus.
>
> **Regarding the use of PWC-Net.**
>
> We kindly clarify that we only use the architecture for PWC-Net without borrowing the pretrained weights. As opposed to low-level dense optical flows, our model learns to predict mid-level segment flows. As shown in Figure 3, our segment flow is less prone to motion noise, articulated motion, or camera motion. This may intrigue new understandings for segmenting objects from videos.
>
> **Accuracy without test-time adaptation**
>
> We use test-time adaptation on video segmentations to exploit the motion information. Test-time adaptation is **not** needed for inference on **static images** such as image saliency detection.
>
> We may run per-frame inference on video object segmentation without test-time adaption. This delivers a mIoU of 47.0 on the validation set without using any motion information, compared with 58.1 of using video motion.  Please note that this performance gap demonstrates the strength of our method, as only an unsupervised learning method on videos such as ours can continue learning during testing.   We will include this result and discussion to the paper. We will add this number to the paper.
>
> **Figure 3 shows significant improvements w.r.t. CIS, yet in Tab. 1 CIS actually performs better?**
>
> Tab 1 provides quantitative evaluations on the full DAVIS validation set, while Figure 3 shows 6 sampled videos. Due to the limited resolution of our motion branch, our current model has a problem with handling two validation videos which contain small objects. These two videos negatively affect the overall performance quite a lot. The problem could be mitigated by designing better architectures or using large resolutions for inference. We have included all our results on all three datasets in the appendix.
>
> **What dataset do the methods in Tab. 2 use for training?**
>
> We use the **same model pretrained on Youtube-VOS** for saliency detection (Tab 2) and video object segmentation (Tab 1). Pretraining details can be found in Sec 3.6 and at the beginning of Sec 4.
>
>
> Thanks for pointing out typos and notations! We will fix them as suggested.

---

> > ### Comment · Reviewer_zbkz · 2021-08-14
> > **Model selection details**
> >
> > Dear authors,
> >
> > thank you for providing the response.
> > I'm curious whether any ground-truth annotation from YouTube-VOS was used to
> > 1) tune the hyperparameter $C$ and
> > 2) to select the channel (class) index of the dominant object in the classification layer of the network.
> > If not, please, provide more details on what empirical measures were used. Does the channel index change between training runs with different random seeds?
> >
> > Thank you for confirming my understanding that a) $C$ and the index of the dominant object are model hyperparameters and remain fixed throughout the inference process; and b) the PWC-Net is initialised from scratch.
> >
> > My comment about Fig. 3 relates to the observation that the qualitative results do not truthfully reflect the quantitative improvement (on DAVIS-2016).

---

> > > ### Author Response · Authors · 2021-08-16
> > > **To Reviewer zbkz**
> > >
> > > Dear Reviewer zbkz,
> > >
> > > Thank you for reading our responses and raising additional questions.
> > >
> > > The ground-truth annotation from Youtube-VOS is never used anywhere across this paper. After pretraining, the channel index is selected as the one whose segmentation mask is closest to the image center, aggregated from all pretraining videos. We also find that this channel could be easily identified as the consistent channel with maximum segment motion. Yes, this channel index changes between training runs with different random seeds.
> > >
> > > Regarding the issue with Fig. 3, examples in the figure are picked to show the difference between traditional dense flow and our segment flow. We admit that the strength of our method is amplified in Fig 3. In the future revision, we will include our failure cases to correct such misunderstandings.  The supplementary materials contain all our testing results on three VOS datasets, which we hope may address the reviewer’s concern for our overall performance. We also append detailed comparisons against CIS for each video on DAVIS 2016 as follows.
> > >
> > > | |blackswan|bmx-trees|breakdance|camel|car-roundabout|car-shadow|cows|dance-twirl|dog|drift-chicane|
> > > |:----|:----|:----|:----|:----|:----|:----|:----|:----|:----|:----|
> > > |AMD(ours)|0.659|0.149|0.613|0.699|0.681|0.786|0.761|0.693|0.628|0.565|
> > > |CIS|0.631|0.57|0.619|0.591|0.854|0.742|0.642|0.723|0.714|0.162|
> > > | |**drift-straight**|**goat**|**horsejump-high**|**kite-surf**|**libby**|**motocross-jump**|**paragliding-launch**|**parkour**|**scooter-black**|**soapbox**|
> > > |AMD(ours)|0.441|0.744|0.562|0.294|0.176|0.368|0.515|0.626|0.53|0.626|
> > > |CIS|0.691|0.224|0.569|0.218|0.64|0.511|0.487|0.68|0.77|0.712|

---

> > ### Comment · Reviewer_zbkz · 2021-08-28
> > **Training data**
> >
> > Thank you for the clarification.
> >
> > The reported results are from training the model on YouTube-VOS, which is substantially larger than the training splits of DAVIS, SegTrack and FBMS used in previous work for learning (almost 100-fold in terms of the number of videos). I wonder if the proposed approach was trained in the previous settings (on DAVIS / SegTrack / FBMS, without  YouTube-VOS), and how fairness of the presented comparison can be justified in view of such discrepancy in the training data.

---

> > > ### Author Response · Authors · 2021-08-30
> > > **To Reviewer zbkz**
> > >
> > > Dear reviewer,
> > >
> > > Thank you for reading our reply and proposing an additional question.
> > >
> > > In previous work CIS and concurrent work MG, they utilize an additional pre-trained flow network (e.g. PWC-Net or RAFT), which is trained on the combination of synthetic datasets FlyingChairs and FlyingThings with flow ground-truth. FlyingChairs dataset contains 22,782 image pairs and FlyingThings provides 39,000 stereo images in a high resolution of 960x540. In contrast, our method does not exploit these datasets and flow supervisions to train our model. Therefore, we believe that it is difficult to strictly align the dataset dependency with prior works.
> > >
> > > The training set of DAVIS only contains 30 videos with 2,079 frames, which are far too small to be considered as an unsupervised pre-training dataset alone. Moreover, DAVIS is well-curated with human screened videos containing a single distinct object. In contrast, Youtube-VOS videos are crawled from Youtube in the wild. Thus, we believe that demonstrating the first work of self-supervised segmentations from videos in the wild at scale is a nice contribution.
> > >
> > > Following the reviewer’s suggestion, we provide additional results for our model trained on DAVIS train set. It obtains an mIoU of 52.0 compared with 58.1 when trained on Youtube-VOS. The saliency result for this model achieves 41.8 on F score, which is a lot worse than 60.2, when trained on Youtube-VOS. Our model is able to exploit the scale and diversity of data for learning.

---

> > > > ### Comment · Reviewer_zbkz · 2021-08-31
> > > > **Why not pre-train on synthetic data**
> > > >
> > > > Dear authors,
> > > >
> > > > thank you again for the clarification.
> > > >
> > > > I understand that the comparison is difficult as previous methods also use synthetic data. However, this configuration seems possible in the proposed framework: why not pre-train the motion net on the synthetic data and fine-tune the framework on the smaller sets of real videos (DAVIS/FBMS/SegTrack)?
> > > > After all, the task of the motion net is, if not identical, closely akin to optical flow estimation.
> > > >
> > > > Otherwise, the advantage of using YouTube-VOS (almost 100K frames) while comparing to previous work seems overwhelming.

---

> > > > > ### Author Response · Authors · 2021-09-01
> > > > > **To Reviewer zbkz: Why not pre-train on synthetic data**
> > > > >
> > > > > Dear reviewer,
> > > > >
> > > > > Thanks for your follow up question.
> > > > >
> > > > > The flow network of the baseline method CIS uses the training data of 60K frames from the combined synthetic datasets of FlyingChairs and FlyingThings. The scale of the synthetic datasets and YouTube-VOS are similar.
> > > > >
> > > > > We agree with the reviewer that pretraining on the synthetic datasets might address the data fairness issue. However, this also brings up an additional issue on whether we should exploit the flow supervision as well, though such supervisions could be obtained freely. It is clear that these flow supervisions are critical for training a high performance dense correspondence network.
> > > > >
> > > > > We wish to clarify that the goal of the paper is **NOT** to propose new techniques to exploit flow supervisions from synthetic datasets. Rather, our goal is to study unsupervised learning from **unlabeled videos**, and transfer the model for downstream segmentation tasks. We believe that it is an exciting direction to exploit large-scale unlabeled videos for enabling downstream applications. And this is under explored by previous approaches. This is the primary reason that we choose not to utilize synthetic datasets.
> > > > >
> > > > > We also wish to emphasize that our model could be transferred to applications on static images while the baseline methods cannot, since they require a dense flow representation as input. Our model achieves such an advantage regardless of the scale of training data.

---

> > > > > ### Author Response · Authors · 2021-09-03
> > > > > **CIS trained on Youtube-VOS**
> > > > >
> > > > > Dear reviewer,
> > > > >
> > > > > We feel that we are unable to provide our results on the synthetic optical flow datasets. Our paper studies unsupervised learning from videos instead of supervised transfer learning from synthetic datasets. The two problems are distinct. Moreover, the FlyingThings dataset is heavily engineered for optical flow with stereo pairs. It is an unrealistic dataset for learning self-supervised video models.
> > > > >
> > > > > To bridge the dataset gap, we propose an alternative solution. For the baseline method CIS, we use the pretrained flow network as before, and we perform segmentation training on the Youtube-VOS dataset, as opposed to the smaller DAVIS dataset. This aligns the training data between CIS and our method. Using its official released code, **CIS trained on Youtube-VOS** only achieves a mIoU of 37.0 compared with 59.1 on the DAVIS validation set. Such decreased performance is due to the uncurated nature of Youtube-VOS videos, facing challenges of large displacement, motion blur, heavy occlusions. This makes the PWC-Net hard to derivate high-quality flow maps, which are critical for CIS. Another reason could be that CIS generator cannot handle multiple objects from the flow input, which are prevalent in Youtube-VOS.
> > > > >
> > > > > We hope that this may address the concern about data fairness.

---

### Official Review · Reviewer_Y1LA · 2021-07-17

**Rating:** 5
**Confidence:** 3

**Summary:**

This paper presents a method that segments the image into a number of segments and estimates an offset vector for each segment. Different from prior two-stream appearance+motion based methods, this approach does not leverage a pre-trained flow network but learns to estimate motion offset vectors implicitly by warping (segmented) regions across frames based on the flow estimates. Reconstruction error (inspired by self-supervised optical flow methods) is used as a loss function and provides a supervisory signal. Thus, the proposed method can be trained using unlabelled video.

**Ethical Concerns:**

I see no ethical concernes.

**Limitations And Societal Impact:**

I see no issues with potential negative social impact, but the limitations were not addressed adequately (details in my review above).

**Main Review:**

Pros
-----------

* This paper is tackling a very relevant research problem of learning to segment objects without supervision, leveraging a large video corpus. The way to go!
* The proposed method is simple but interesting and with the potential to scale very well.
* The paper is generally well-written, and the methodology is presented clearly.
* Results do indeed look quite promising: on-par with state-of-the-art on DAVIS, compared to methods that use no supervision.
* I appreciate that the paper makes an effort to demonstrate the versatility of this method for several tasks, ranging from VOS to object segmentation in images and as unsupervised pre-training for semantic segmentation.


Cons
-----------

* This paper claims that this method is trained absolutely without any supervision. In principle, this is true: network weights are initialized randomly before training the model on YouTube VOS dataset. However, I would be careful regarding “no supervision” claims because the results demonstrated were obtained via training on a highly curated dataset containing various biases, most notably photographer bias (objects are almost always centered in these videos). This suspicion is pretty much confirmed in L182-184: “We have empirically observed that the moving objects all appear in a particular mask channel across the training videos. This channel can be heuristically identified as the one whose segmentation mask is closest to the image center.” It, therefore, appears to me that the model learns to segment object that appears in the center of the video (or image), based on carefully selected videos that mostly contain one object of interest per clip (objects are present from start to the end of the video). This fundamentally differs from real “in-the-wild” scenarios (e.g., learning from a stream of sensory data that a robot observes). On the other hand, let me also note that most papers that claim to do unsupervised learning do not acknowledge the manual human effort invested in curating datasets that are used for learning. To conclude this thought, the paper would severely benefit from clearly stating assumptions made and limitations.
* It is not clear to me at all how ResNet50 backbone + a FCN head yield segmentation of an image into multiple segments. Here a crucial piece of information is missing.
* Optical flow methods (that several baselines use) can be trained in a self-supervised fashion, or using synthetic data. It can be argued that these methods do not use supervision either. I am not saying that it is not “nice” to be able to train the model from scratch, not relying on any external components, but given that baselines (CIS, MG) obtain better or significantly better results on DAVIS, this point should be argued better.
* While Tab. 1 shows promising results, results and conclusions are also somewhat confusing. Apart from traditional methods, both learning-based baselines perform better on DAVIS (they do, though, use optical flow, but see my comment above). The proposed method performs better than CIS on SegTrack and FBMS, but those two datasets contain stronger photographer bias (correct me on this if I am mistaken).
* I also have my doubts regarding zero-shot detection results on the DUTS dataset, and I am not sure if I agree with the conclusion. It again seems to me that the method learned to detect exactly the object that is right in the center (looking at images, Fig. 4). A better version of this experiment would be per-frame evaluation on DAVIS, which also contains sequences where the object is not always centered, and occasionally multiple objects are present.
* I also have concerns regarding using this approach as unsupervised pre-training (which in general is a great idea!). First of all, I would definitely need to see ImageNet pre-training for comparison purposes, otherwise, it is difficult to anchor the results reported wrt. prior work. The same goes for SimSiam and MoCo-v2: ImageNet-trained variants are missing. I do understand why the paper reports these two methods trained on the same data (which indeed should be done in addition to ImageNet trained variants), but I got extremely confused with augmentations here. The core idea behind contrastive learning-based methods such as SimSiam and MoCo is to leverage augmentations to form positive pairs. If those are not used, I do not understand how these methods are trained. Perhaps I misunderstood something.

Tips
-----------

* This paper advertises zero-shot segmentation, but I don't think the terminology is used accurately (wrt. Xian et al., Zero-Shot Learning -- The Good, the Bad and the Ugly, CVPR’17) as the classes that are segmented in the DUTS dataset are present during the training phase, at least most of them, however, they are not labeled. This is rather a case of transductive zero-shot learning. This doesn’t impact my rating, just a remark/tip.


Justification
-----------

In principle, I like the idea behind this paper: it is simple but has the potential to be quite effective, and I believe that there is a need for such methods for unsupervised representation learning/object segmentation (simple methods that can scale). However, at the moment, I see too many issues with this manuscript to recommend acceptance at this time. The most critical part is the experimental evaluation: in general, I like that the paper makes an attempt to demonstrate the versatility of this approach for several tasks (VOS, object segmentation, unsupervised pre-training). However, no experiment is particularly convincing (see my criticism above), and raises more questions than provides answers.

I would recommend the following:
* Provide a more thorough discussion on the assumptions and limitations of this approach (see my comments on photographer bias and dataset curation).
* It would be interesting to see a variant of the proposed method that does leverage optical flow as input (wrt. Tab. 1). However, this experiment is the least problematic in my view.
* More importantly, claims regarding zero-shot saliency detection and unsupervised pre-training need stronger experimental verification; in particular, detection capabilities should be evaluated in settings where the object is not always centered and potentially multiple objects are present. Otherwise, the experimental setting is too constrained. Furthermore, I cannot judge the performance of this approach for unsupervised pre-training because the results are not discussed and placed well relative to prior work (see my comments above). Ideally, unsupervised pre-training should also be compared to prior work on unsupervised representation learning from video (the first one that comes to mind is Wang et al., Unsupervised Learning of Visual Representations using Videos, CVPR’14).

Other than that, thumbs up for generally well-written paper and method that appears to me to hold quite a lot of potential!

**Post rebuttal comments and justification**
It was particularly challenging for me to reach my final recommendation for this paper; as I've mentioned, I like this research direction. The simple yet scalable approach to learning representations from abundant video looks very appealing to me, and I find this work valuable.

In the end, the main question was how convincing is the experimental evaluation and how firm conclusions I can make. As mentioned in my review, I do appreciate that evaluation is versatile, but in the end, I am afraid that I still think that none of the experiments is highly convincing.

VOS evaluation Reviewer zbkz raises the concern that those comparisons are not entirely fair, as the proposed method is the only one to train on YouTube-VOS, which is for order of magnitude larger than DAVIS, SegTrack, and FBMS. Baselines (CIS, MG) were only trained on the target datasets but did utilize pre-trained flow networks. I am afraid I agree that the fairest direct comparison would be to pre-train the network using synthetic data (commonly used to train optical flow networks) and fine-tune it on DAVIS/FBMS/SegTrack.

But perhaps the real question would be: how useful such a pre-trained network would be for downstream tasks, for example, semantic segmentation?

It turns out that this method significantly lags behind SOTA (MoCo-v2, SimSiam, 72.4 mIoU) on VOC2012. I agree with the assessment that ImageNet is better aligned with VOC2012. However, (i) the task itself (moving object segmentation) is more similar to the target task of semantic segmentation, and (ii) there are large automotive datasets out there to which this method could be applied (e.g., pre-train on BDD100k [Yu et al., CVPR'19], train on Cityscapes). I am afraid that the paper did not succeed yet to demonstrate its merit in pre-training for semantic segmentation.

I am afraid I am also not convinced with the results obtained on the DUTS dataset; it appears to me that the network learns to detect anything in the center. I am afraid I disagree with the author's statement that this center bias is present everywhere (it isn't present in robotics/automotive datasets). Another way to show the merit would be to use this approach for pre-training on the instance segmentation task (e.g., on COCO).

I would very be happy to see a future revision of this paper that would present a very convincing experimental evaluation. I would point to [Radford et al., ICLR'21 (CLIP)] for reference (simple method, thorough evaluation) and entirely focus on demonstrating the merit via convincing experimental evaluation.

**Time Spent Reviewing:**

4h

---

> ### Author Response · Authors · 2021-08-10
> **To Reviewer Y1LA**
>
> Thank you for the detailed comments. These constructive comments would help us greatly improve the quality of the paper.
>
> **This paper claims that this method is trained absolutely without any supervision.**
>
> Our method is unsupervised in the sense that it does not explicitly use human annotated segmentations, just as unsupervised learning or self-supervised learning papers conduct experiments on curated datasets such as ImageNet or Kinetics.  We agree with the reviewer that such a dataset implicitly contains knowledge and we will provide clarifications in the paper.
>
> **Dataset bias, especially the center bias problem of the YouTube-VOS dataset.**
>
> Perhaps this confusion stems from how **we choose the segment channel** and further gets mixed up with **the potential dataset bias issue**. We will first address the latter here and then the channel selection next.
>
> The Youtube-VOS dataset is a video segmentation dataset collected as a subset of the Youtube-8m dataset. During construction, it uses key words or phrases to query the Youtube  videos.  There is **no careful  human selection or post-processing of videos** where objects are centered. In fact, we observe that Youtube-VOS videos are very challenging, often exhibiting very fast and large motion, severe occlusions, object disappearances, and reappearances.
>
> We agree with the reviewer on the photographer bias that moving objects may be systematically biased towards the center of the image frame. However, our method of appearance and motion decomposition does not have any built-in prior of object locations in the images. That is, our model is generic:  It neither assumes nor relies on the center locality of moving objects.
>
> **Further explanations on choosing object segments from $C$ channels.**
>
> After pretraining, we empirically find that there is only one segmentation channel (1 of the 5 channels of model weights) that corresponds to the foreground object, and that this channel is consistent across downstream tasks.  It is important to note that we do not adjust this channel per testing image or video. We choose this segment channel from the training set, and we apply this channel for all testing cases.
>
> **Arguments for end-to-end training over dense  flow-based methods.**
>
> Our approach goes beyond low-level dense optical flows towards **mid-level segment flows**.  This conceptual jump could inspire a new line of methods for segmenting objects from videos.  Our approach also delivers significant practical gains:  It could be directly applied to single images during test time, whereas dense flow-based methods cannot.  In addition, our end-to-end formulation simplifies model pipeline engineering and helps facilitate future development.
>
> **How an image is segmented into multiple segments using ResNet50 + an FCN head.**
>
> The segmentation network follows the common practice used in semantic segmentation. For instance, an image of size $ 384 \times 384 \times 3$ is fed into ResNet50 for a spatial feature of dimension $48\times48\times2048$.  A  fully convolutional network head containing  2 convolution blocks processes the feature into $48\times48\times C$​  spatial softmax predictions, where $C$​ is the number of segmentation maps. We will clarify this detail in the paper.
>
> **Performance evaluations with CIS on DAVIS, SegTrack-v2 and FBMS59.**
>
> Our model performs 1% worse than CIS on DAVIS, and 12% to 15% better on SegTrack-v2 and FBMS59 respectively. These two datasets are in fact more challenging than DAVIS, noticing that the absolute mIoU performance on SegTrack-v2 and FBMS59 is in fact worse than on  DAVIS. Each sequence on SegTrack-v2 contains 1-6 moving objects, presenting challenges of motion blur, complex deformation, interacting objects, occlusion and so on. Sequences on FBMS59  also involve multiple objects. We have included our results of all videos in the appendix. In many videos of SegTrack and FBMS, the object is not in the center or there are multiple objects, e.g. 2s, 15s, 19s in fbms.mp4 and 5s, 10s, 16s in segtrack.mp4.
>
> **Zero-shot detection results on the DUTS dataset benefits from dataset bias.**
>
>  DUTS is a widely used saliency benchmark dataset. Our model trained on Youtube-VOS  videos  is directly applied to DUTS images **without tuning a single parameter.**  Thus, our method remains generic without any specific tuning to the bias in the testing set.
> We agree with the reviewer that DUTS images may exhibit photographer bias which places objects in the center.  However, such an inductive bias is widely present in photos, not just in the DUTS images. In Figure 4, row 1 column 3, our model can detect horses regardless of its position in the image.  Our method does not benefit from the specific test set distributions on DUTS.
>
> **Transfer learning baselines MoCo and SimSiam w/ and w/o augmentations.**
>
> MoCo-v2 and SimSiam transfer results with ImageNet pretraining achieves 72.4, outperforming our result of 62.0 by a large margin. This gap has two major causes:  1) ImageNet with 1M images is much larger and diverse than Youtube-VOS with 3500 videos; 2) ImageNet is well aligned with VOC2012 on semantic object classes.  We thus compare our method with contrastive pretraining on the same Youtube-VOS data in Table 3.  We plan to add ImageNet pretraining results of MoCo and SimSiam in the revision with these discussions.
>
> Following the reviewer’s advice, we have compared with a video representation learning baseline *“Wang et al., Learning correspondence from the cycle-consistency of time, CVPR 2019”.* Using the released pretrained model, it obtains a mIoU of 52.8, which is much lower than our method of 62.0.
>
> Sorry about the confusion from our imprecise description on augmentations! In fact, we remove heavy augmentations such as color jittering and gray-scaling while maintaining light augmentations of standard spatial cropping to study the augmentation effects. Our method does not require heavy augmentations to achieve better transfer results. This will be revised accordingly.
>
> Thanks very much for the tip regarding zero-shot learning :-) We will fix minor issues as suggested.

---

> ### Author Response · Authors · 2021-08-31
> **To Reviewer Y1LA**
>
> Dear reviewer,
>
> Thank you very much for your comments.
>
> We hope our reply could address your concerns. If you have any other comments, please contact us and we are very pleased to clarify as the end of discussion is approaching.
>
> Thank you for your effort and time!

---

> ### Comment · Reviewer_Y1LA · 2021-09-02
> **Thanks for the feedback!**
>
> Dear authors,
>
> Thanks for the feedback and clarifications!
>
> It was particularly challenging for me to reach my final recommendation for this paper; as I've mentioned, I like this research direction. The simple yet scalable approach to learning representations from abundant video looks very appealing to me, and I find this work valuable.
>
> In the end, the main question was how convincing is the experimental evaluation and how firm conclusions I can make. As mentioned in mt review, I do appreciate that evaluation is versatile, but in the end, I am afraid that I still think that none of the experiments is highly convincing.
>
> **VOS evaluation** Reviewer zbkz raises the concern that those comparisons are not entirely fair, as the proposed method is the only one to train on YouTube-VOS, which is for order of magnitude larger than DAVIS, SegTrack, and FBMS. Baselines (CIS, MG) were only trained on the target datasets but did utilize pre-trained flow networks.
> I am afraid I agree that the fairest direct comparison would be to pre-train the network using synthetic data (commonly used to train optical flow networks) and fine-tune it on DAVIS/FBMS/SegTrack.
>
>
> **But perhaps the real question would be:** how useful such a pre-trained network would be for downstream tasks, for example, semantic segmentation?
>
> It turns out that this method significantly lags behind SOTA (MoCo-v2, SimSiam, 72.4 mIoU) on VOC2012. I agree with the assessment that ImageNet is better aligned with VOC2012. However, (i) the task itself (moving object segmentation) is more similar to the target task of semantic segmentation, and (ii) there are large automotive datasets out there to which this method could be applied (e.g., pre-train on BDD100k [Yu et al., CVPR'19], train on Cityscapes). I am afraid that the paper did not succeed yet to demonstrate its merit in pre-training for semantic segmentation.
>
>
> I am afraid I am also not convinced with the results obtained on the DUTS dataset; it appears to me that the network learns to detect anything in the center. I am afraid I disagree with the author's statement that this center bias is present everywhere (it isn't present in robotics/automotive datasets). Another way to show the merit would be to use this approach for pre-training on the instance segmentation task (e.g., on COCO).
>
> I would very be happy to see a future revision of this paper that would present a very convincing experimental evaluation. I would point to [Radford et al., ICLR'21 (CLIP)] for reference (simple method, thorough evaluation) and entirely focus on demonstrating the merit via convincing experimental evaluation.

---

> > ### Author Response · Authors · 2021-09-03
> > **To Reviewer Y1LA**
> >
> > Dear reviewer,
> >
> > We thank the reviewer for taking time for reading our responses, providing positive assessments and additional concerns. We wish to have the opportunity to address the concerns regarding empirical evaluations.
> >
> > **VOS evaluation.** Our approach is not intended for exploiting optical flow ground truth for pretraining. Using synthetic images alone while neglecting these flow supervisions puts our approach in an obvious disadvantageous position. Instead, our approach is intended for unsupervised learning from videos. In our opinion, learning from synthetic data with ground truth labels and learning from unlabeled videos are two related but distinct problems. Given that  the reviewer acknowledges the direction of unsupervised learning from abundant videos, we hope that the reviewer may also agree that supervised pretraining from synthetic datasets deviate from this direction. We kindly refer the reviewer to our response for reviewer zbkz for additional information.
> >
> > **DUTS evaluation.** DUTS dataset (http://saliencydetection.net/duts/) is the largest saliency benchmark, widely recognized in the image saliency research community. Hundreds of papers perform evaluation on this dataset [1], and research progress is reflected by the metric on this dataset. We believe that our result on this benchmark would be recognized in the saliency research community. The reviewer’s concern basically challenges the fundamental value of the DUTS dataset and saliency detection research. The suggestion on extending our method to instance segmentation is highly non-trivial and clearly beyond the scope of the paper.
> >
> > **Semantic segmentation evaluation.** We admit that our model does not reach the state-of-the-art results for semantic segmentation. However, we do show that our model is very promising when comparing with our baselines, i.e. contrastive models trained on the same video data and a recent video representation learning work Cycle-Time [2]. The results against our baselines are strong and convincing.
> >
> > **Dicussions on CLIP.** We share our fondness for the CLIP paper. Its strong empirical evidence is demonstrated through large-scale training, which is beyond typical research labs. We wish to scale our model on very large datasets with large model capacity in the future. However, this is beyond the scope of the paper.
> >
> > [1] Wang, L., Lu, H., Wang, Y., Feng, M., Wang, D., Yin, B., & Ruan, X., Learning to Detect Salient Objects with Image-level Supervision, CVPR 2017
> >
> > [2] Wang, X., Jabri, A., & Efros, A. A. Learning correspondence from the cycle-consistency of time. CVPR 2019

---

### Official Review · Reviewer_pZ87 · 2021-07-17

**Rating:** 6
**Confidence:** 4

**Summary:**

Paper proposes a novel approach to unsupervised object segmenation based on temporal consistency in video sequences.
The authors show good results across a range of video datasets, and that the models trained without supervision generalise well to figure/ground segmenations on image based datasets.

The method itself is conceptually straightforward, decomposing the image into a fixed number of parts so that each part has a low photometric error when propegated to a future frame, using a constant velocity model.

**Limitations And Societal Impact:**

Fine. Work is unlikely to have direct social impact.

 Visualisation of the other segments that do not correspond to objects would be welcome, and help the reader understand the behaviour/limitations better, as would showing some failure cases.

**Main Review:**

Paper is well written and the results are compelling, clearly outperforming unsupervised approaches, and compairing well with approaches that make use of supervised components.

The experiments are extensive, and the inclusion of supervised methods that perform better than this approach is welcome, and helps put the overall results in context.

Arguably this paper would be better suited for a CV conference, essentially we have very standard ML choices composed with a standard loss, and limitted novelty from a pure ML perspective. However, I believe that the task itself - completely unsupervised object discovery from video is of strong interest to the wider ML community, and the paper therefore is a good fit for neurips.

I'd characterise it as having sufficient novelty, that's made up for by the relevance of the approach to the community.

Minor comments:

Line 177 Don't use FCN as a standalone abreviation, it can stand for either fully connected network, or fully convolutional network, and leaves the reader guessing.

non-published -> unpublished


**Time Spent Reviewing:**

5 hours

---

> ### Author Response · Authors · 2021-08-10
> **To Reviewer pZ87**
>
> Thank you for the positive feedback of the paper.
>
> **Visualization of the  other segments and showing some failure cases.**
>
> We observe that the other segments which do not correspond to the objects are often unstructured and not very meaningful, e.g., a mixture of static road and sky. These intuitive results indeed facilitate understanding and we will include them in the revised version. For unsupervised video segmentation, we have attached all testing results on three datasets in the appendix, including failure cases. We will append more saliency detection results especially for failure cases.
>
>
> **Limitations of our approach.**
>
> Our approach uses a simple constant velocity assumption to model the motion within each segment. This may be a limitation when the motion within the object segment is extremely complex. Also, our approach models motion between adjacent frames without considering the long-term pattern of a moving object.  These limitations can be addressed in future work within our two-stream decomposition pipeline with more advanced components.
>
>
>
>
> We will fix the typos and other minor issues as suggested.

---

> ### Author Response · Authors · 2021-08-31
> **To Reviewer pZ87**
>
> Dear reviewer,
>
> Thank you very much for your comments.
>
> We hope our reply could address your concerns. If you have any other comments, please contact us and we are very pleased to clarify as the end of discussion is approaching.
>
> Thank you for your effort and time!

---

### Official Review · Reviewer_5AXj · 2021-07-20

**Rating:** 5
**Confidence:** 4

**Summary:**

This paper proposes to decouple the motion and appearance by two pathways: the appearance pathway is designed to learn object segmentation masks, while the motion pathway aims to predict the optical flow per region mask. The whole model is learned in a self-supervised manner by frame-reconstruction loss. While such region-wise, which is in contrast to the typical way of learning pixel-wise dense optical flow, is interesting, the overall novelty is not sufficient for NeurIPS.

**Limitations And Societal Impact:**

The whole framework sounds feasible, however, the model is designed coarsely and I doubt whether the full potential of such framework has been exploited by such implementation. For instance, the appearance pathway is designed simply by multi-convolutional layers to predict 'c' masks, which is arduous to learn the segmentations effectively. The motion pathway obtains the region-wise optical flow simply by pooling the predicted pixel-wise motion features. The whole model is supervised by simple frame-reconstruction loss. I doubt its effectiveness. As shown in the paper, the model can only identify the primary object in a scene where there is only one primary moving object. Besides, the strategy for identifying the primary mask by  "This channel can be heuristically identified as the one whose segmentation mask is closest to the image center" is not an elegant solution.

**Main Review:**

Originality: The idea that learning the region-wise optical flow to identify primary/moving objects is relatively new. The framework of decoupling the motion and appearance features by two single pathways is not quite novel. Overall, the originality is limited.

Quality: The overall framework of the submission is technically sound. However, since the detailed implementations of each modules (two pathways) are too simple, I doubt its effectiveness.  I doubt that such implementation under the self-supervised learning can only simply segment the primary object.

Clarity: Overall the submission is written clearly.

Significance: the novelty is limited.




**Time Spent Reviewing:**

10 hours

---

> ### Author Response · Authors · 2021-08-10
> **To Reviewer 5AXj**
>
> Thank you for reviewing our paper. However, it is unfortunate that your major concerns, on novelty and effectiveness respectively, are too brief and unsubstantiated for us to provide further clarification.
>
> **1)** *“The framework of decoupling the motion and appearance features by two single pathways is not quite novel. Overall, the originality is limited”*.
>
> Could you provide any references to published works, for we could find none?
>
> **2)** *“Since the detailed implementations of each module (two pathways) are too simple, I doubt its effectiveness”*.
>
> Yes, we study this new important topic with a simple model; however, the effectiveness is justified not by the complexity, but by our extensive experimental validation on a wide range of tasks. If a simple model can get the job done, why not?
>
> We urge the reviewer to consider our work more carefully and provide concrete justifications for any concerns.

---

> ### Author Response · Authors · 2021-09-01
> **To Reviewer 5AXj**
>
> Dear reviewer,
>
> Thank you very much for your comments.
>
> We hope our reply could address your concerns. If you have any other comments, please contact us and we are very pleased to clarify as the end of discussion is approaching.
>
> Thank you for your effort and time!

---

### Decision · Program_Chairs · 2021-09-27

**Decision:**

Accept (Poster)

**Comment:**

Reviewers did not reach a consensus for this paper. Most of them like the idea in the paper and consider that the experimental section is comprehensive, but some of the reviewers are not fully convinced by any of the experiments.

There was also considerable discussion over if the method depends strongly on photographer bias. I am not too concerned about that in training: one can imagine a curriculum of learning where a method like the proposed is first exposed to videos with photographic bias, and progressively it moves to more complex data. Arguably the proposed model may not scale all the way, also since it has simplifying assumptions like a constant velocity motion model, but it may be able to serve as a good bootstrapping mechanism. There was some criticism in discussion that the model trains on more data than competing methods, but being unsupervised i wouldn't hold that against the method

I join the reviewers in enjoying the simplicity of the approach and understand that the experiments can be improved but think that they are decent already. In any case i encourage the authors to look carefully into the reviews and improve the experiments following the reviewers advice, namely trying to understand how good the segments are in multi-object data. I would be curious if the model still learns on more complex video data, such as imagenet-video or larger action recognition datasets.